# Discovery of re-purposed drugs that slow SARS-CoV-2 replication in human cells

Adam Pickard[1,2]*, Ben C. Calverley[1,2], Joan Chang[1,2], Richa Garva[1,2], Sara Gago[2], Yinhui Lu[1,2], Karl E. Kadler[1,2]*

1 Wellcome Centre for Cell-Matrix Research, University of Manchester, Oxford Road, Manchester, United Kingdom, 2 School of Biological Sciences, Faculty of Biology, Medicine & Health, University of Manchester, Manchester Academic Health Science Centre, Manchester, United Kingdom

* adam.pickard@manchester.ac.uk (AP); karl.kadler@manchester.ac.uk (KEK)

**Data Availability Statement:** All relevant data are within the manuscript and its Supporting Information files.

## Abstract

COVID-19 vaccines based on the Spike protein of SARS-CoV-2 have been developed that appear to be largely successful in stopping infection. However, therapeutics that can help manage the disease are still required until immunity has been achieved globally. The identification of repurposed drugs that stop SARS-CoV-2 replication could have enormous utility in stemming the disease. Here, using a nano-luciferase tagged version of the virus (SARS-CoV-2-ΔOrf7a-NLuc) to quantitate viral load, we evaluated a range of human cell types for their ability to be infected and support replication of the virus, and performed a screen of 1971 FDA-approved drugs. Hepatocytes, kidney glomerulus, and proximal tubule cells were particularly effective in supporting SARS-CoV-2 replication, which is in-line with reported proteinuria and liver damage in patients with COVID-19. Using the nano-luciferase as a measure of virus replication we identified 35 drugs that reduced replication in Vero cells and human hepatocytes when treated prior to SARS-CoV-2 infection and found amodiaquine, atovaquone, bedaquiline, ebastine, LY2835219, manidipine, panobinostat, and vitamin D3 to be effective in slowing SARS-CoV-2 replication in human cells when used to treat infected cells. In conclusion, our study has identified strong candidates for drug repurposing, which could prove powerful additions to the treatment of COVID.

## Author summary

The identification of repurposed drugs that stop SARS-CoV-2 replication could have enormous utility in stemming COVID-19. Here, using a nano-luciferase tagged version of the virus (SARS-CoV-2-ΔOrf7a-NLuc) to quantitate viral load, we evaluated a range of human cell types for their ability to be infected and support replication of the virus, and performed a screen of 1971 FDA-approved drugs. Hepatocytes, kidney glomerulus, and proximal tubule cells were particularly effective in supporting SARS-CoV-2 replication, which is in-line with reported proteinuria and liver damage in patients with COVID-19. Using the nano-luciferase as a measure of virus replication we identified 35 drugs that reduced replication in Vero cells and human hepatocytes when treated prior to SARS-CoV-2 infection and found amodiaquine, atovaquone, bedaquiline, ebastine, LY2835219,

**Funding:** The research was funded by Wellcome (London) (110126/Z/15/Z and 203128/Z/16/Z) to KEK. SG was funded by the NIHR Manchester Research Centre and the Fungal Infection Trust. The funders had no role in study design, data collection and analysis, decision to publish, or preparation of the manuscript.

**Competing interests:** The authors have declared that no competing interests exist.

manidipine, panobinostat, and vitamin D3 to be effective in slowing SARS-CoV-2 replication in human cells when used to treat infected cells. In conclusion, our study has identified strong candidates for drug repurposing, which could prove powerful additions to the treatment of COVID.

## Introduction

The COVID-19 pandemic caused by the severe acute respiratory syndrome coronavirus 2 (SARS-CoV-2) virus is having a widespread impact on global health with substantial loss of life. SARS-CoV-2 infection in patients with COVID-19 can result in pulmonary distress, inflammation, and tropism to multiple organs [1]. Some infected individuals are asymptomatic whilst others mount an exaggerated immune response, or 'cytokine storm'[2,3], which correlates with multiple organ failure and poor outcome [4]. Vaccines have been developed to help protect people from COVID-19 but the emergence of mutational variants, some with increased transmissibility, calls for additional research into alternative treatment and prevention strategies.

An early step in the infection process is interaction of the Spike protein on the surface of the virus with angiotensin-converting enzyme 2 (ACE2) on the surface of the host cell [5]. Using this information, first-generation vaccines have been generated against the Spike protein. SARS-CoV-2 entry can be potentiated by additional host factors including neuropilin-1 [6]. Once inside, the virus uses components of the host cell to replicate and secrete viral particles [7] and disrupt RNA handling and protein translation to suppress host defenses [8]. The different stages of the disease, from the initial infection of host cells through to virus replication and the response (normal or extreme) of the immune system, offer opportunities to identify drugs, treatments, and therapies to help stop disease progression. Furthermore, large proportions of the world's population remain at risk of contracting COVID-19 as they wait to be vaccinated, which has prompted governments to encourage people to continue to self-isolate, remain at home, and maintain social distancing. The identification of safe and easily distributed medications that can target the different stages of virus infection and replication, could reduce the spread of SARS-CoV-2 and reduce the cases of COVID-19.

The Food and Drug Administration (FDA) and the European Medicines Agency (EMA) work with pharmaceutical companies to develop safe and effective drugs for the benefit of public health. Using a library of 1971 FDA-approved compounds, here we used a traceable clone of SARS-Cov-2 to identify well-characterized drugs that could potentially slow the infection and replication of the SARS-CoV-2 virus in human cells.

We had previously developed an approach to quantitate collagen synthesis by using CRISPR-Cas9 to insert the gene encoding NanoLuciferase (NLuc) into the Col1a2 gene in fibroblasts [9]. Here, we adapted the approach to tag the SARS-CoV-2 virus with NLuc, and then used the recombinant NLuc-tagged virus to monitor virus replication and to identify drugs that slow the replication process. SARS-CoV-2 was originally recovered by culturing human specimens in the African green monkey kidney cell line, Vero [10], and the virus readily replicates in this cell type. We have performed initial screening in the Vero cell line, and to draw reliable conclusions from our studies we performed a replicate screen in a human hepatocyte cell line, HUH7. We evaluated a panel of human cell types for their ability to be infected by SARS-CoV-2 and to support virus replication, and optimized culture conditions for each cell type to provide a robust system for screening in human cells. We found that hepatocytes

and kidney epithelial cells were proficient in supporting SARS-CoV-2 replication whereas fibroblasts and lung epithelial cells supported only minimal replication.

Therapeutic compound screens have been performed using Vero cells infected with SARS-CoV-2 [11–13], cell viability or presence of viral proteins have been used to evaluate effectiveness of each therapeutic. Similar screens have also been performed in human cell types [12,14], however many of the identified compounds are found to suppress virus infection [11]. Using a nano-luciferase activity as a marker of virus replication we have evaluated how 1971 FDA-approved compounds impact on SARS-CoV-2 replication, in Vero and HUH7 cells. Through applying stringent criteria in our hit selection and confirming efficacy of these compounds in suppressing replication in cells already infected with SARS-CoV-2, we identified a shortlist of nine drugs that were effective in inhibiting SARS-CoV-2 replication. The drugs identified included anti-cancer and anti-viral drugs, but also fewer toxic drugs such as atovoquone, ebastine and vitamin D3. The multiple steps involved in virus infection and proliferation, means that each of these drugs may have different molecular targets, and for some, additional targets are still being elucidated. As these drugs are FDA-approved and with safe dosimetry already established for use in patients, clinical trials could be initiated for these drugs within a relatively short time frame.

## Results

### Generation of a traceable SARS-CoV-2 virus

High throughput screens have been developed to identify drugs suitable for re-purposing for treatment of COVID19 (e.g., [11,15]). These screens have been performed in non-human cell lines, such as Vero, and rely on secondary factors such as cell viability to identify candidates, or markers that indicate virus infection. To monitor the replication of SARS-CoV-2 we have generated a modified traceable virus where Orf7a is replaced with the sequence encoding NanoLuciferase (SARS-Cov-2-ΔOrf7a-NLuc, **Fig 1A**). NanoLuciferase (NLuc) is an enzyme that produces light when supplied with its substrate (coelenterazine) and is readily detectable even at low quantities [9]. Orf7a has previously been removed in SARS-CoV and SARS-CoV-2 and yielded infectious and replicative virus particles, in order to accommodate the NLuc sequence we followed these designs [16–18]. We have used a reverse-genetics approach to generate virus particles (**S1 Fig**), and plaque forming assays demonstrated that the replication of SARS-CoV-2-ΔOrf7a-NLuc viruses were equivalent to the wild-type SARS-CoV-2 (Wuhan-Hu-1, NC_045512.2) (**Fig 1B**). Viral RNAs (**S1 Fig**), and electron microscopy (**Figs 1C, 1D and S2**) confirmed virus production. NLuc activity was readily detected in infected cultures upon addition of the substrate coelentrazine (**Fig 1E**).

### NLuc activity as a marker of virus replication

To monitor replication of the SARS-CoV-2-ΔOrf7a-NLuc virus, Vero cells were exposed to increasing numbers of replicative virus particles (plaque forming units, PFU) and luminescence was used as a measure of virus replication. Luminescence measurements were compared to SARS-CoV-2-ΔOrf7a-NLuc virus in the absence of cells as background (**Fig 2A**), 24, 48 and 72 hours post infection (h.p.i.). A lag in virus replication of at least 48 hrs was observed but replication was readily observed at 72 hrs, when as little as 2 PFU per well (MOI 0.0004) was added. Using these optimization experiments 72 h.p.i. was chosen as the time point at which to assess virus replication in subsequent assays, the average Z' factor [19] for our replication assay in 96 well format was 0.8 indicating suitability for library screening. Amplification of NLuc signal upon virus infection was consistent for all passages of the traceable virus (**S3 Fig**) and with varying cell number (**S4 Fig**).

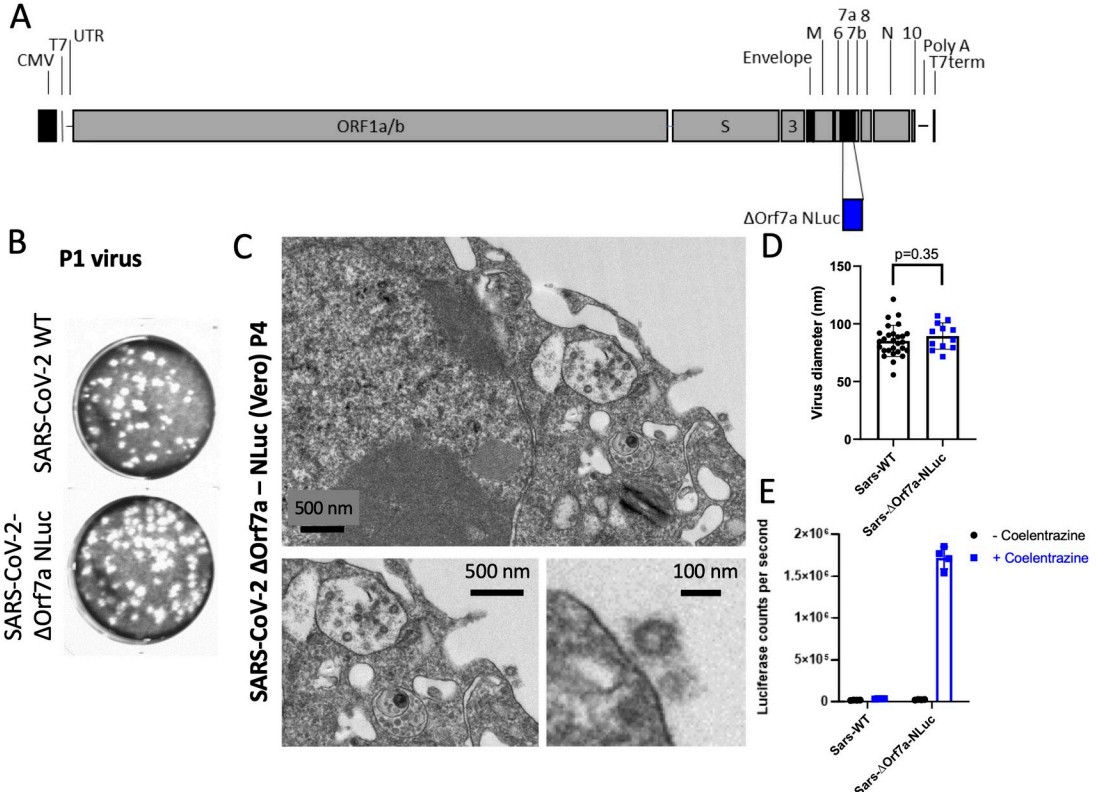

**Fig 1. Nanoluciferase (NLuc) modified SARS-CoV-2 virus as a reporter for virus replication.** A) Diagram of the SARS-CoV-2 genome highlighting the insert site for the reporter NLuc in place of Orf7a, (SARS-CoV-2-ΔOrf7a-NLuc). B) Virus particles recovered following transfection of wild-type (WT) or NLuc modified SARS-CoV-2 encoding RNA into 293T cells were used to infect Vero cells. The recovered virus (P1 virus) was then titered in Vero cells to assess virus replication and plaque forming potential. C) Electron microscopy of the SARS-Cov-2 virus 72 h.p.i. of Vero cells (Passage 4, P4), both intra- and extra-cellular virus particles were identified. D) Measurement of diameter for the wild-type (WT) and NLuc modified SARS-CoV-2 particles. E) Measurement of NLuc activity in medium of Vero cells infected wild-type (WT) and NLuc modified SARS-CoV-2 virus without and with addition of the NLuc substrate coelentrazine.

## SARS-CoV-2 replication screen validation

High throughput screens have been developed to identify drugs suitable for re-purposing for treatment of COVID19 (e.g., [11,13,20]). These screens have been performed in non-human cell lines, such as Vero, and rely on secondary factors such as cell viability to identify candidates. Other screening methods have used detection of viral proteins and have been able to identify candidates that impact virus infection [12,14]. A first step in validating our viral replication screen was to measure virus replication after treatment with interferon alpha 2 (IFNA2) which has previously demonstrated to reduce virus replication [17]. When Vero cells were infected with a single PFU, pre-treatment of IFNA2 was able to reduce replication of SARS-CoV-2-ΔOrf7a-NLuc in a dose dependent manner. However, with higher virus load (100 PFU), IFNA2 had little effect on virus replication (**Fig 2C and 2D**). These results provided confidence that the NLuc-tagged virus system could be used to quantify the effects of drugs on virus replication, especially if reduction of bioluminescence was detected with high PFU.

## Drug repurposing to target SARS-CoV-2 replication

A 1971 FDA-approved compound library was used to pre-treat Vero cells before infection with either 1PFU (MOI 0.0002) or 100 PFU (MOI 0.02) SARS-CoV-2-ΔOrf7a-NLuc virus as

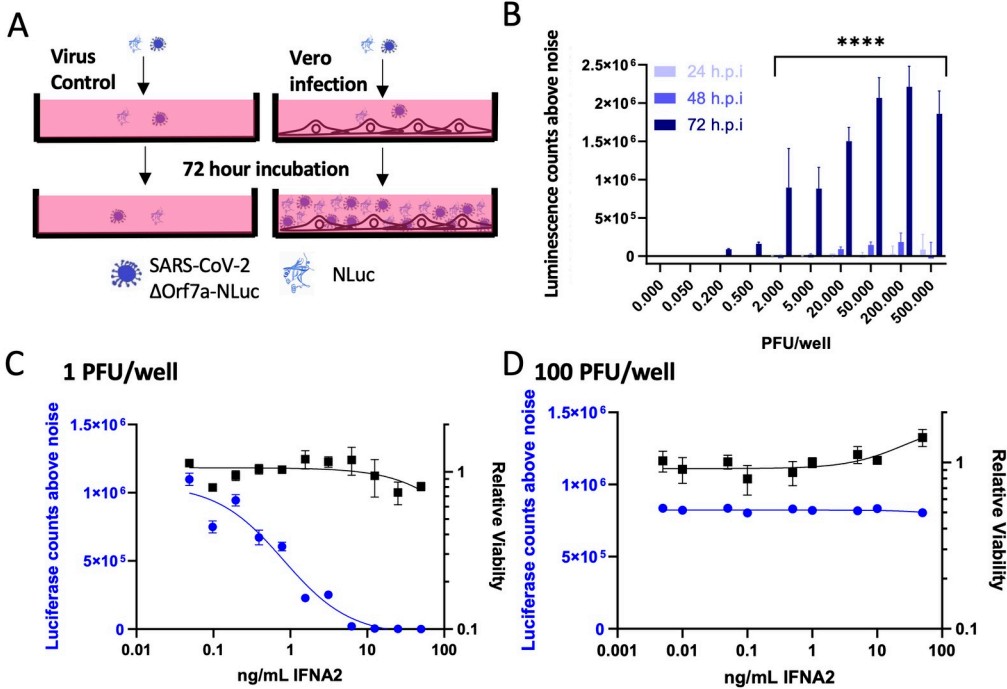

**Fig 2. Timings of SARS-CoV-2-ΔOrf7a-NLuc virus replication.** A) Schematic showing how SARS-CoV-2 ΔOrf7a-NLuc was used to assess virus replication. During virus replication as virus particles are released from the cell, or because of cell death, NLuc activity is detected in the conditioned medium together with virus particles. In subsequent assays the NLuc activity is recorded in the absence of cells, and NLuc signals above background signal are used to indicate the level of virus replication. B) The replication of SARS-CoV-2-ΔOrf7a-Nluc was monitored in a 96 well assay. Using 5000 Vero cells per well, increasing numbers of virus particles were added and monitored 24, 48 and 72 hrs post infection (h.p.i). Substantial and significant increases in NLuc activity were observed when only two plaque forming units (PFU) were added. Higher NLuc activity was associated with higher viral input. Even with higher viral inputs the substantial increases in NLuc activity occurred only after 72 h.p.i. C) Five thousand Vero cells were pre-treated with increasing doses of IFNα2 for 24 hrs prior to infection with 1PFU SARS-CoV-2-ΔOrf7a-NLuc. NLuc activity and viability were assessed 72 h.p.i. demonstrating effective inhibition of virus replication. The blue line indicates the luminescence counts per second for DMSO treated controls. N = 3 replicate samples. D) As in C, 5000 Vero cells were treated with increasing doses of IFNα2 for 24 hours prior to infection with 100 PFU SARS-CoV-2-ΔOrf7a-NLuc. With this higher titer, virus replication was not inhibited. The blue line indicates the luminescence counts per second for DMSO treated controls. N = 3 replicate samples.

outlined in **Fig 3A**. Infected Vero cells that were untreated and background (virus only wells) were used as controls. The Z' factor was assessed for each plate of the screen averaging 0.867± 0.140 SD and 0.915± 0.058 SD, for the 1PFU and 100PFU screen, respectively demonstrating a robust assay. More hits were identified in the 1PFU screen and therefore we focused our hit selection on compounds that reduced virus replication by at least 85% in the 100PFU screen (**Fig 3B** and **S1 Table**). We identified 69 compounds that reduced replication (FDR<0.1 compared to untreated and DMSO controls, 55 showed no viral replication). These hits were also identified in the duplicate screen performed with 1PFU per well (**Fig 3C**). Five of these compounds were false positive hits due to direct inhibition of NLuc activity (**Fig 3C** and **S2 Table**).

## Drug repurposing screen for SARS-CoV-2 replication in human cells

Having established an effective screening strategy in Vero cells, we sought to establish a similar approach in a human cell line capable of supporting SARS-CoV-2 replication. SARS-CoV-2 has been reported to infect multiple cell lines, and replication of the viral genome has been identified [21]. We tested the replicative capacity of SARS-CoV-2-ΔOrf7a in the lung epithelial

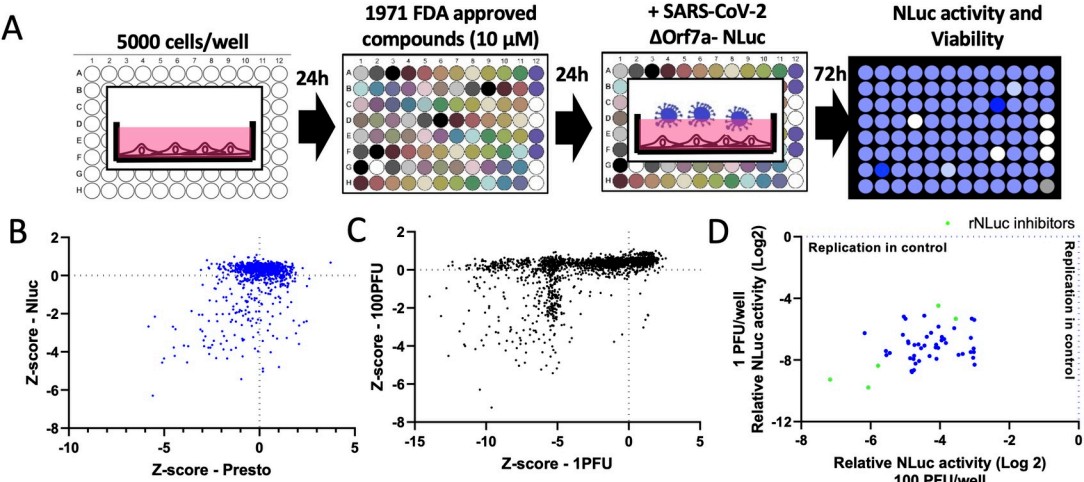

**Fig 3. Screen of 1971 FDA-approved compounds to identify therapeutics that inhibit SARS-CoV-2-ΔOrf7a-NLuc virus replication.** A) Schematic of the screening procedure used to assess whether FDA-approved compounds alter SARS-CoV-2 ΔOrf7a-NLuc virus replication. B) Scatterplot showing the effects of 1971 compounds on SARS-CoV-2 ΔOrf7a-NLuc replication and cell viability. C) Scatterplot showing the effects of 1971 compounds on SARS-CoV-2 ΔOrf7a-NLuc replication in replicate screens using 100 PFU or 1 PFU per well. The Z-score for luciferase activity is plotted. D) Scatterplot of 50 compounds that reduced NLuc activity in C. The luciferase activities relative to untreated controls (indicated by the blue dashed line) are shown. Highlighted in green are compounds that inhibit the NLuc enzyme directly.

cell lines Calu3, A549, and 16HBEo and in lung fibroblasts. Using real-time PCR and western blotting for the SARS-CoV-2 nucleocapsid protein, we observed virus replication in these cell lines, and we confirmed that SARS-CoV-2 WT and SARS-CoV-2-ΔOrf7a-NLuc viruses infected these lung cell lines (**S5 Fig**). However, the Z' factor suggested that they were not suitable for high throughput screening (**Fig 4A**), this was also observed for primary lung epithelial cells (**S6 Fig**). We assessed additional cell lines reported to be infected by SARS-CoV-2 including Caco-2 [20,21] and HUH7 [10], as well as additional kidney cell lines. These were chosen as we were able to recover virus particles from 293T cells and kidney is reported to express high levels of ACE2 [22]. Replication in monocytic cells and fibroblastic cell lines was also monitored due to the systemic and fibrotic nature of COVID-19. We identified virus replication in HUH7 cells, microvascular cells of the kidney glomerulus, and proximal tubule cells of the kidney and THP1 monocytic cells (**Fig 4A**).

As each cell line has a specific culture medium for optimal growth, we set out to test if different media would support cell growth and virus replication in lung epithelial cells, as this is the primary site of Sars-CoV-2 infection. We used 15 different medium conditions to grow lung epithelial cells prior to infection with SARS-CoV-2-ΔOrf7a-NLuc; for comparison, HUH7 cells were also included. Changes in growth conditions improved assay performance in lung epithelial and fibroblasts but signal-to-noise ratios remained low. In contrast, for HUH7 cells, these optimized conditions significantly improved the 96 well format assay as indicated by Z' factor >0.5 and signal-to-noise ratios >20 (**Fig 4B and 4C**). Using the 1971 FDA-compound library we identified 223 compounds that suppressed SARS-CoV-2 replication by greater than 85% whilst maintaining cell viability. We refined these hits by overlapping with positive hits from the screen in Vero cells (**Fig 4E and S3 Table**). We identified 35 inhibitory compounds and 2 compounds that increased NLuc activity, that were common to the two screens (**Fig 4F**). The intended clinical use or targets of the 35 inhibitory compounds included anti-virals, antibiotics, modifiers of dopamine and estrogen receptor activity, calcium ion channel inhibitors and HMG-CoA reductase inhibitors (**Fig 5A**). The hits also identified

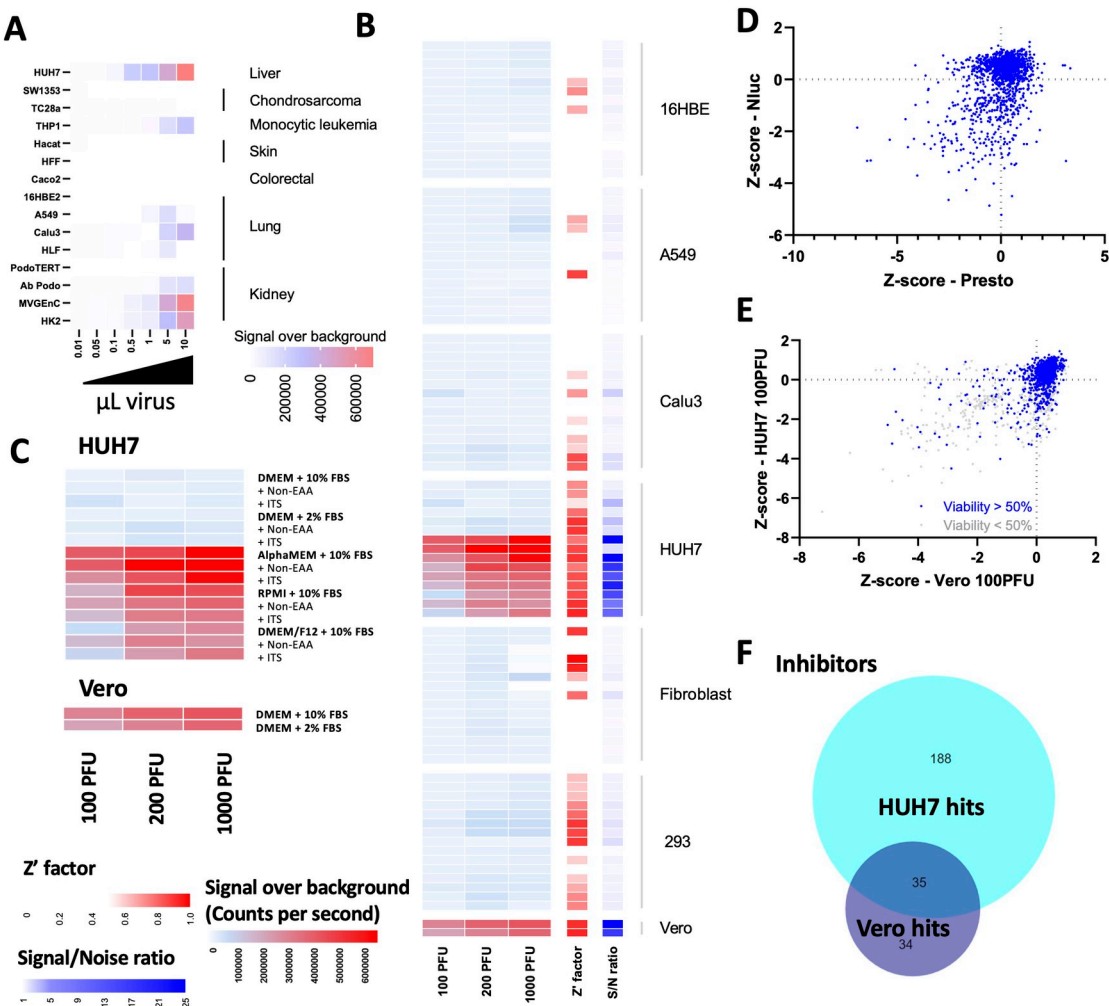

**Fig 4. Replication of SARS-CoV-2-ΔOrf7a-NLuc in human cell lines.** A) SARS-CoV-2 replication in human cell lines. Five thousand cells per well were seeded and infected as in **Fig 3A**. All cells were grown in the recommended growth medium, detailed in **Table 1**. The NLuc activity after 3 days was used to assess the degree of virus replication. NLuc activity above the baseline activity is shown as a heat map. Each box is the average of 2 biological repeats of each cell line at each virus dose, with the same findings observed when 2000, or 10,000 cells were seeded. B) Lung epithelial, fibroblasts, HUH7 and 293T cell lines were grown in 15 different growth medium compositions. NLuc activity over background three days after infection is shown. Each row represents a different growth medium as indicated in C). Each box represents 4 replicate measures of replication. Similar findings were observed in n = 6 biological repeats. Z' factors were calculated for each virus dose using 'virus only' wells as negative controls, the average Z' factor for each growth medium is shown. The signal-to-noise ratio of infected cells relative to 'virus only' wells are also shown. C) NLuc activity above background three days after infection in HUH7 cells. Each box represents 4 replicate measures of replication. Similar findings were observed in n = 6 biological repeats. D) The effects of 1971 FDA-approved compounds on the replication of SARS-CoV-2-ΔOrf7a-NLuc (100 PFU/well, MOI 0.02) in HUH7 cells grown in alpha-MEM supplemented with 10% FBS. Five thousand cells were treated with 10 μM of each compound for 24 hrs prior to infection with 100 PFU of virus. Virus replication progressed for 72 hrs as in **Fig 3A**. E) Comparison of compounds identified by screening for HUH7 and Vero cells Z-scores for luciferase activity are shown. Comparison and overlap of inhibitors identified in the compound screening for HUH7 and Vero cells.

vitamin D3, which is available over the counter. Multiple vitamin D related compounds were present in the screen and these too suppressed SARS-CoV-2-ΔOrf7a-NLuc although these did not all meet our criteria for both Vero and human cell lines (**Fig 5B**).

These initial hits were further tested to determine if treatment could also *prevent* replication in cells already infected with SARS-CoV-2-ΔOrf7a-NLuc. We infected Vero cells with

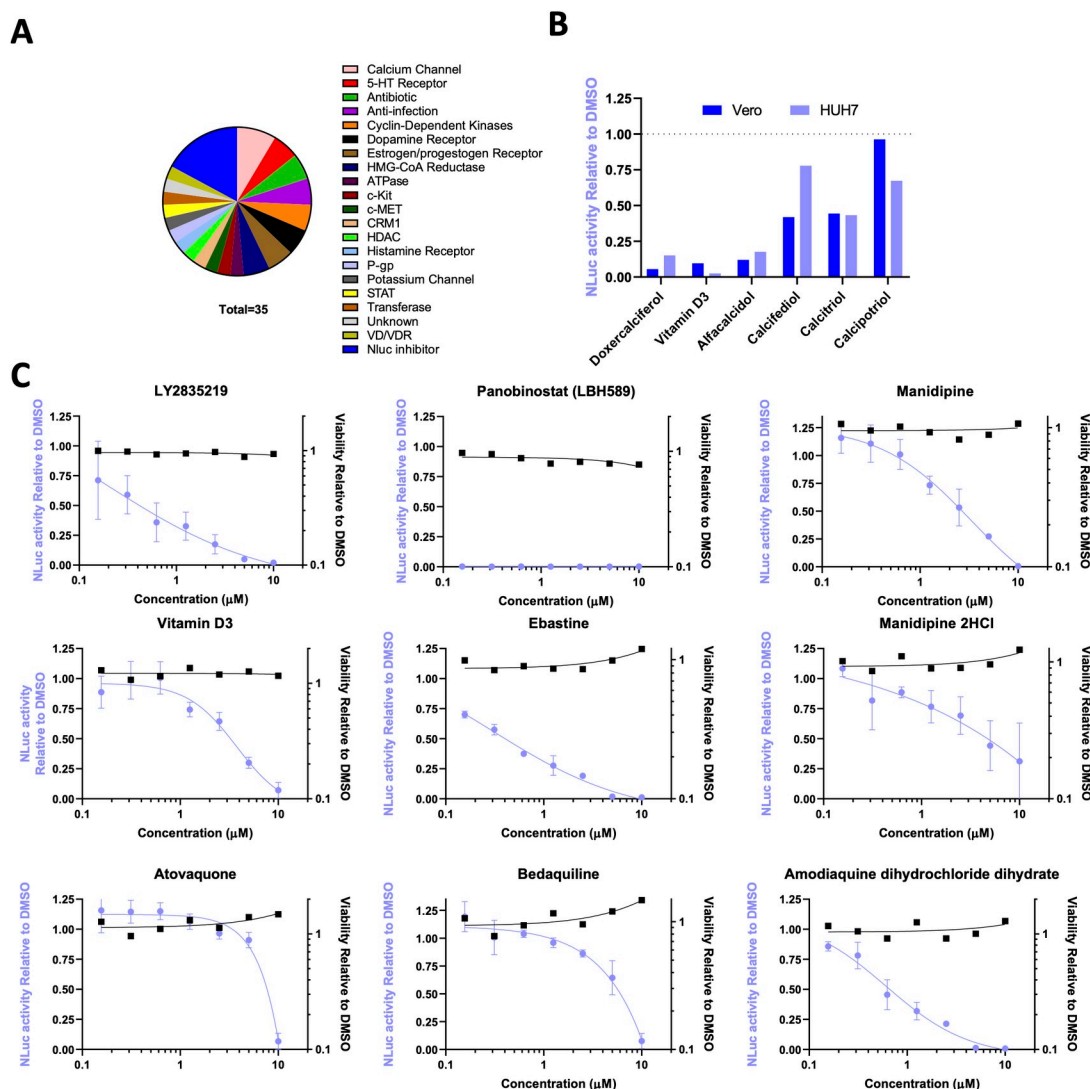

**Fig 5. Dose response of SARS-CoV-2 inhibitors.** A) Summary of the inhibitor class of the 35 compounds identified in both the Vero and HUH7 screens. B) Effects of vitamin D related compounds from the APExBIO DiscoveryProbe library on NLuc activity in the Vero and HUH7 screens. C) Dose responses to the 9 compounds in HUH7 cells. Cells were treated and infected as described in **Fig 4D**. NLuc activity relative to DMSO controls are shown in blue, and viability relative to DMSO are shown in black. N = 2 independent repeats are shown.

SARS-CoV-2 for 24 hrs (which is sufficient time for the virus to begin replication) then added each of the 35 inhibitor compounds. The cells were incubated for 48 hrs (i.e. a total of 72 h.p.i) prior to bioluminescence being measured. Most of the compounds had no impact on virus replication; but 9 of the 35 compounds reduced replication relative to DMSO controls (**Table 1**, **S7 Fig**). The effective doses for these compounds were then determined in HUH7 cells to provide a future reference for selecting dose after comparison to pharmacokinetic data established in human trials (**Fig 5**, **Table 1**). We also tested the ability of these 9 compounds to inhibit the replication of the wild-type SARS-CoV-2 virus in HUH7 cells, all compounds reduced virus titers (**Fig 6**). Whilst these are pre-clinical *in vitro* data, they demonstrate the efficacy of the compounds in reducing virus replication post infection and warrant further investigation to determine if these could ease the burden of the virus in patients.

**Table 1. Approved uses of the 9 compounds of interest that inhibit SARS-CoV-2 infection and replication in HUH7 cells.**

| Compound | HUH7 IC50 | Vero IC50 | Post-infection at IC50 | Approved Use | Target |
|---|---|---|---|---|---|
| Panobinostat | <0.2 μM | <0.2 μM | -25% | Multiple myeloma | HDAC inhibitor |
| LY2835219 | 0.2 μM | 1 μM | -40% | Abemaciclib for advanced breast cancer | CDK4/6 inhibitor |
| Manidipine | 2 μM | 7.5 μM | -50% | Anti-hypertensive | Calcium channel blocker |
| Manidipine 2HCl | 2.5 μM | 7.5 μM | -50% | Anti-hypertensive | Calcium channel blocker |
| Ebastine | 0.5 μM | 5 μM | -25% | Anti-histamine for allergic rhinitis and chronic idiopathic urticaria | H₁ receptor blocker |
| Atovaquone | 7.5 μM | 3 μM | -25% | antimicrobial (anti-malarial) | Selective inhibitor of parasite mitochondrial electron transport |
| Bedaquiline | 5 μM | 10 μM | -30% | pulmonary tuberculosis | mycobacterial ATP synthase |
| Vitamin D3 | 3 μM | 10 μM | -30% | Common supplement | Transcriptional/calcium regulation |
| Amodiaquine | 1 μM | 5 μM | -25% | antimicrobial (anti-malarial) | Inhibitor heme polymerase activity |

## Discussion

In this study we have shown that the SARS-CoV-2 virus infects and replicates in a range of human cells especially hepatocytes, kidney glomerulus, and proximal tubule cells of the kidney, and, that 9 drugs that have previously been shown to be safe in humans and approved by the

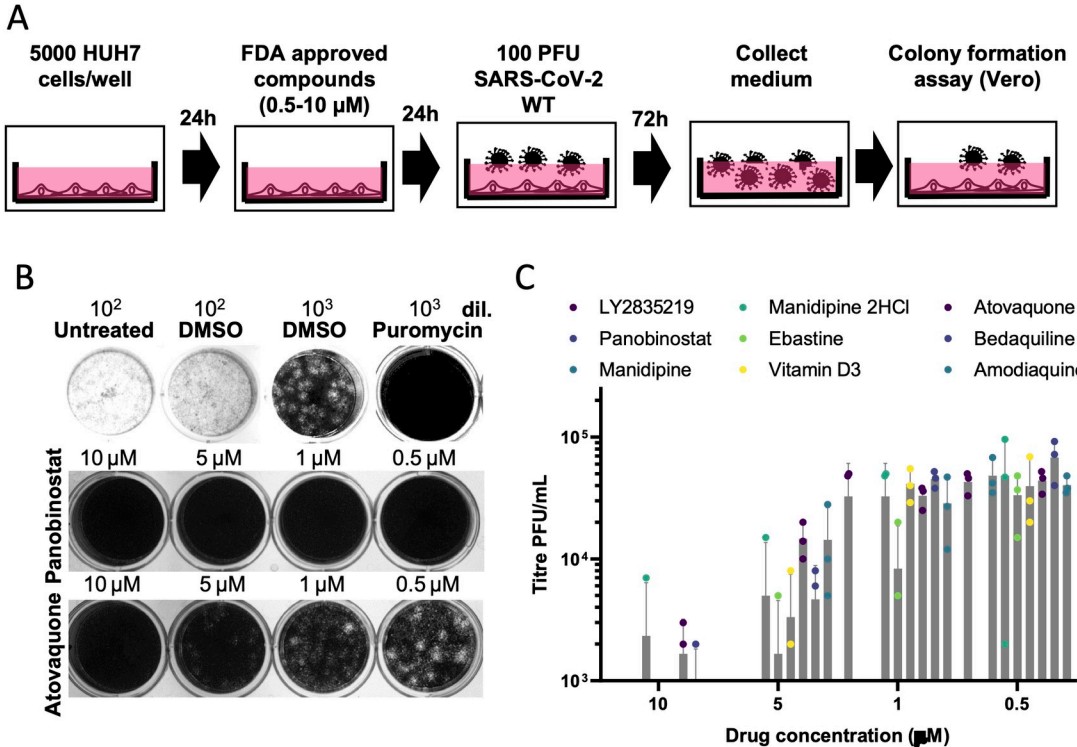

**Fig 6. Effect of inhibitors on wild-type SARS-CoV-2 replication.** A) SARS-CoV-2 replication in HUH7 cells treated with identified inhibitors was assessed by titering virus-containing medium. B) Example plaques obtained using medium from control (untreated or DMSO treated) or treated cultures of HUH7 cells infected with wild-type 100 PFU SARS-CoV-2. Medium was diluted 1000-fold to assess numbers of plaque forming units in treated samples. C) Wild-type virus titers from HUH7 cells treated with varying doses of the 9 compound hits. N = 3 independent repeats are shown, error bars represent the standard deviation.

FDA for clinical use are effective in inhibiting SARS-CoV-2 replication even after infection. We have demonstrated that, whilst we were able to slow virus replication with many more compounds if cells were pretreated prior to infection, many of these compounds failed to impact on virus replication if applied after infection. This finding is supported by observations of a large compound library screen in Vero cells which demonstrated that many of the identified compounds suppressed virus uptake [11].

The identification that liver and kidney cells are infection and replication competent for SARS-CoV-2 aligns with the observed liver and kidney abnormalities in patients with COVID-19. Liver comorbidities have been reported in 2–11% of patients with COVID-19, and 14–53% of cases reported abnormal levels of liver enzymes [23], with liver injury being more prevalent in severe cases (reviewed by [24]). Another study reported that patients with chronic liver disease, especially African Americans, were at increased risk of COVID-19 [25]. It has been suggested that liver damage in patients with COVID-19 might be caused by viral infection of liver cells, which is supported by the presence of SARS-CoV-2 RNA in stool [26]. On a similar note, acute kidney injury is a common complication of COVID-19 and has been associated with increased morbidity and mortality (reviewed by [27]). Our data showing SARS-CoV-2 infection and replication in kidney cells helps to explain kidney pathology associated with COVID-19.

RECOVERY [28] has set out to identify therapeutics that could be re-purposed for treatment of COVID patients. For example, dexamethasone, which is a broad spectrum immunosuppressor, has been applied clinically to inhibit the destructive effects of the cytokine storm and shown to reduce mortality and decrease the length of hospitalization [29], did not demonstrate any effect on viral replication in our study. It may be possible in the future to combine the therapeutics identified in our study with dexamethasone so that both the cytokine storm and virus replication are targeted.

Of interest, other therapeutics that have been tested in the RECOVERY trial include ritonavir and lopinavir (antiretroviral protease inhibitors), which failed to show clinical benefit in the treatment of COVID-19 [30]. These drugs were present in the DiscoveryProbe library used in our study but showed no significant effect on SARS-CoV-2 replication, in our assays. Anti-viral agents such as Ritonavir and Lopinavir, which failed to show effects in the RECOVERY/AGILE trials, also failed to halt replication of SARS-CoV-2 in either our Vero or HUH7 cell screens. Other trialled compounds such as hydroxychloroquine and azithromycin did show some effects in HUH7 cells but these effects were largely due to their toxic effects on cells.

The high costs and lengthy lead-in times associated with new drug development, make repurposing of existing drugs for the treatment of common and rare diseases an increasingly attractive idea (for review see [31]). The approaches used include hypothesis driven, preclinical trials including computational (e.g., computational molecular docking) and experimental (e.g., biochemical or cell-based assays of drug interactions) methods, and evaluation of efficacy in phase II clinical trials. As noted by Pushpakom *et al.*, of these three steps, the identification of the right drug for an indication of interest (in our case SARS-CoV-2 and COVID-19) is critical.

The nine drugs that we identified here have been approved for use in the treatment of a variety of diseases. Panobinostat, for example, is a HDAC inhibitor that blocks DNA replication, and has been used to inhibit cell growth in the management of cancer. Panobinostat had the strongest effect on limiting SARS-CoV-2 replication whilst maintaining cell viability, and completely blocked replication of SARS-CoV-2 at all doses tested (**Fig 5**); however, if cells were infected prior to treatment a more modest effect on replication were observed. This difference may be related to the recent observations that panobinostat can suppress ACE2

expression [32], which would explain its beneficial effect prior to entry of the virus into cells and not thereafter. Abemaciclib (LY2835219) is another cell cycle inhibitor, suppressor of DNA replication and anti-cancer drug that emerged from the screen however these are likely to be detrimental to recovery from SARS-CoV-2 infection.

Atovaquone is of particular interest because it has been identified in other studies of SARS-CoV-2 in the context of COVID-19. It is a hydroxynaphthoquinone approved by NICE for the treatment of mild to moderate pneumocystis pneumonia and as a prophylaxis against pneumocystis pneumonia. It also has been used in combination with proguanil (Malarone) as an antimalarial. The mechanism of action of atovaquone has been widely studied; it is a competitive inhibitor of ubiquinol, and against *P. falciparum*, it acts by inhibiting the electron transport chain at the level of the cytochrome bc1 complex [33]. A more recent study showed that atovaquone inhibits Zika and Dengue virus infection by blocking envelop protein-mediated membrane fusion [34]. Furthermore, in silico molecular docking strategies suggested potential binding of atovaquone to the SARS-CoV-2 spike protein [35,36]. These two mechanisms of action help to explain how atovaquone slows both infection and replication of SARS-CoV-2 in cells.

Our observation that bedaquiline inhibits SARS-CoV-2 replication supports evidence from an *in silico* repurposing study in which bedaquiline was proposed to be a promising inhibitor of the main viral protease of SARS-CoV-2 [37]. The main protease cleaves pp1a and pp1ab polypeptides, which encode nonstructural proteins to form the replication-transcription complex [38], and help explain how inhibitors to the main protease are effective in inhibiting replication of SARS-CoV-2 in Vero 76 cells [39]. Manidipine (a calcium ion channel blocker approved for the use in treating hypertension) is proposed to reduce the activity of the main protease of SARS-CoV-2 [40] and limit the availability of calcium ions, which are required for insertion of the coronavirus fusion peptide into the host lipid bilayers during viral fusion [41].

Our screen also identified ebastine (a second generation H1 receptor antagonist that has been approved for the treatment of allergic rhinitis and chronic idiopathic urticaria) and vitamin D3, which is a health supplement available over the counter. However, histamine antagonists exhibit effects in addition to blockage of the histamine receptor. For example, ebastine blocks the release of anti-IgE-induced prostaglandin D2 [42]. The vitamin D receptor (VDR, a member of the nuclear hormone receptor superfamily) is proposed to be essential for liver lipid metabolism because its deficiency in mice protects against hepatosteatosis [43]. Once bound to VDR, vitamin D plays a major role in hepatic pathophysiology, regulation of innate and adaptive immune responses, and might contribute to anti-proliferative, anti-inflammatory and anti-fibrotic outcomes (reviewed by [44]). Thus far, whether vitamin D supplementation reduces the risk of SARS-CoV-2 infection or COVID-19 severity is unclear [45]. Whilst vitamin D3 met the stringent cut-offs of 85% virus reduction, vitamin D2 and other vitamin D related therapeutics also reduced virus replication but did not meet all criteria across both Vero and HUH7 cell lines.

In conclusion, our study has identified compounds that are safe in humans and show effectiveness in reducing SARS-CoV-2 infection and replication in human cells, especially hepatocytes. Their potency in stopping SARS-CoV-2 replicating in human cells in the face of the COVID pandemic, warrants further study.

## Materials and methods

### Cell culture

Cell lines maintained in growth medium are shown in **S4 Table**.

## Generation of functional SARS-CoV-2 virus

DNA encoding the genome of SARS-CoV-2 and SARS-CoV-2-ΔOrf7a-NLuc were purchased from Vectorbuilder Inc. (Chicago, US). Transfection of DNA encoding the viruses failed to generate replicative virus when electroporated into 293T cells. We therefore produced RNA molecules which encoded the virus *in vitro*. Briefly, virus encoding DNA (1 μg) was transcribed using T7 mMessenger mMachine (Thermo) with a GTP:Cap ratio of 2:1 used in a 20 μL reaction. In addition, RNA encoding the SARS-CoV-2 nucleocapsid was also generated by PCR using primers P1 and P2 (**S5 Table**). It has been reported that this aids the recovery of replicative virus [18]. Viral RNA genomes (10 μl) and 2.5 μL nucleocapsid RNA were electroporated into 293T cells within the BSL3 laboratory. Viral RNAs were electroporated (5,000,000 cells, 1100 V, 20 ms and 2 pulses) and grown in T75 cm$^2$ flasks and 24 well plates. Cells grown in 24 well plates for 24–120 hrs were used to monitor changes in NLuc activity.

## Virus production, maintenance and assessment of titer

Culture medium was collected from 293T cells 6 days after electroporation. This virus (P0) was used to infect cells of interest. As virus replication was slow in 293T cells, virus stocks were maintained by passage in Vero cells grown in DMEM supplemented with 2% FBS. Medium (1 mL) was used to infect Vero cells in order to generate P1 virus. Replication was assessed by measuring NLuc activity over 10 days. For subsequent passage of the virus, Vero cells in T75 cm$^2$ flasks were infected with 2 mL of medium containing virus, after 3 days the medium was collected and passed through 0.45 μm filters using Luer-loc syringes.

To titer the virus, 200,000 Vero cells were seeded in 6-well plates overnight in growth medium. After removing growth medium Vero cells were infected with 200 μL of serially diluted virus containing medium at 37˚C. After 1 hour, wells were overlaid with 2 mL of 0.3% low melt agarose in 2x DMEM containing 1% FBS and grown for 3 days. After 3 days infected cells were fixed with 10% PFA overnight and then stained with crystal violet. Plaques were identified by imaging plates on BioDoc-It gel documentation system (UVP, Upland, US).

To detect viral RNA in medium, 0.25 mL of virus containing medium was collected 3 days post infection, 0.75 mL TriPure LS reagent (Sigma-Aldrich St. Louis, US) was added, and RNA isolated according to the manufacturer's recommendations, in the final step RNA was dissolved in 15 μL DNAse/RNAse free water. For detection of SARS-COV-2 nucleocapsid transcripts in lung epithelial cells, 200,000 cells were infected at the indicated MOI for 3 days, monolayers were lysed directly in 1 mL Trizol (Invitrogen, Paisley UK). For assessing expression of ACE2, TMPRSS2 and NLP1 expression in lung epithelial cells RNA was isolated from the cells using Trizol and RNA isolated according to the manufacturer's recommendations. For cDNA generation and real-time PCR we used conditions previously described [46] using primers detailed in **S5 Table**.

## Electron microscopy

After 3 days infection with P4 virus, Vero cells were scrapped and pelleted. Cell pellets were fixed using 2.5% glutaraldehyde and 4% paraformaldehyde in 0.1 M cacodylate buffer for 24 h, washed in ddH$_2$O three times, 30 mins for each wash. Cell pellets were incubated in freshly made 2% (vol/vol) osmium tetroxide and 1.5% (wt/vol) potassium ferrocyanide in cacodylate buffer (100 mM, pH 7.2) for 1 hr at room temperature. Samples were washed in ddH$_2$O five times each for 3 minutes. Specimens were transferred to freshly made 1% (wt/vol) tannic acid in 100 mM cacodylate buffer (pH 7.2) for 40 mins at RT and washed in ddH$_2$O five times for 3 mins each at RT. The specimen was incubated with 1% (vol/vol) osmium tetroxide in ddH$_2$O for 30 minutes at room temperature and washed in ddH$_2$O three times for 5 min each at room

temperature. The specimen was then incubated with 1% (wt/vol) uranyl acetate (aqueous) at 4˚C for 16 hrs (overnight) and then washed in ddH$_2$O three times for 5 mins each time at room temperature.

Specimens were dehydrated in graded ethanol: 30, 50, 70, 90% (vol/vol) ethanol in ddH$_2$O for 10 mins at each step. Then samples were washed four times for 10 mins each time in 100% ethanol at room temperature. Samples were transferred to propylene oxide for 10 mins at room temperature.

The specimen was finally infiltrated in a graded series of Agar100Hard in propylene oxide at room temperature: first for 1 hour in 30% (vol/vol) Agar100Hard, 1 hr in 50% (vol/vol) Agar100Hard then overnight in 75% (vol/vol) Agar100Hard, and then 100% (vol/vol) Agar100Hard for 5 hrs. After transferring samples to freshly made 100% Agar100 Hard in labelled molds and allowed to cure at 60˚C for 72 hrs. Sections were imaged on an FEI Tecnai12 BioTwin.

## NLuc activity assay

Vero cells grown in 24-well plates (Corning, 3526) were assayed for NLuc activity by adding 1 µL of coelenterazine (final concentration 1.5 µM). For 96 well formats, cell lines were seeded in white walled microwell plates (Nunc MicroWell 96-Well, Nunclon Delta-Treated, Flat-Bottom Microplate, Thermo Fisher Scientific, Paisley, UK# 136101). To measure NLuc activity, 0.5 µL coelenterazine was added per well (final concentration 3 µM). Light production was measured using filter cubes #114 and #3 on the Synergy Neo2 Multi-Mode Reader (Biotek), readings for each well were integrated over 200 ms with 4 replicate measurements per well (Gain 135 and read height 6 mm). For viability measurements, 2.5 µL Prestoblue (Thermo Fisher Scientific, Paisley UK) was added per well incubated for 10 mins before reading fluorescence at Excitation: 555/20 nm, Emission: 596/20 nm (Xenon flash, Lamp energy low, Gain 100 and read height 4.5 mm, 10 measurements per data point).

## Drug screens

The DiscoveryProbe FDA-approved library of 1971 compounds (L1021, APExBIO Boston, US) was prepared as follows. After thawing the library for 4 hours at room temperature the library was arrayed into 96 well plates at 1 mM in DMSO and stored at -20˚C. Stocks (1 mM) were thawed at room temperature for 2 hrs before compounds were added to cells.

For all drug screens and validation, 5000 cells were seeded in 50 µL of growth medium for 24 hrs in white walled microwell plates. DMEM containing 2% FBS was used for Vero cells and alpha-MEM containing 10% FBS was used for HUH7 cells. The following day 0.5 µL of each compound (final concentration 10 µM) was added per well and incubated for 24 hours. Eighty-eight compounds were tested per plate and each plate contained the following controls: two untreated wells, two DMSO treated wells, and one well treated with 10 µM puromycin to kill cells. SARS-CoV-2-ΔOrf7a-NLuc virus was added to all wells. In addition, two wells did not contain any cells but were infected with the SARS-CoV-2-ΔOrf7a-NLuc virus as a measure of background NLuc activity. A final well which contained cells but were uninfected were also included. Twenty-four hours after drug treatment, cells were infected with 100 PFU per well SARS-CoV-2-ΔOrf7a-NLuc virus in 50 µL of the indicated growth medium for 72 hrs. To assess virus replication and viability, 2.5 µL of Prestoblue was added to each well and plates incubated for 10 mins at 37˚C. Coelenterazine was then added to a final concentration of 3 µM. Plates were sealed prior to reading luciferase activity and viability as described above. Validation of hits were performed using the same procedures described for the drug screen.

## Study design and statistical analysis

To identify hit compounds, raw luciferase luminescence reads ($x$) were normalized relative to the virus-infected drug-untreated controls ($u$) and plate minimum read ($m$) on each plate by the formula:

$$x_{norm} = \frac{x_{raw} - m}{u - m}$$

This meant that a normalized luciferase value of 1 implied no difference from untreated virus replication, and a value of 0 represented total inhibition of viral replication.

The PrestoBlue reads ($p$) were normalised relative to the virus-infected DMSO-treated controls ($d$) and plate minimum read ($n$) on each plate by the formula:

$$p_{norm} = \frac{p_{raw} - n}{d - n}$$

This meant that a normalized PrestoBlue value of 1 implied no difference in cell viability from DMSO-treated virus infected cells, and a value of 0 representing maximal reduction in cell viability.

Z-scores were calculated relative to the mean $\log_2$(fold change) for each plate.

Compounds were categorized as either inhibitors or enhancers of NLuc-SARS-CoV-2 activity. Inhibitors were compounds where normalized NLuc-SARS-CoV-2 levels ($x_{norm}$) was reduced such that $x_{norm} < 0.15$, and cell viability as measured by normalised PrestoBlue levels ($p_{norm}$) wasn't affected by more than 50%, such that $p_{norm} > 0.5$.

Where indicated one-tailed Student's T-test were performed to evaluate significance in changes to NLuc activity.

## Supporting information

**S1 Fig. Recovery of replication competent SARS-CoV-2 from synthetic DNA constructs.**
A) Schematic for the recovery of SARS-CoV-2 virus particles from DNA encoding the wild type and NLuc modified SARS-CoV-2 genome. RNA was transcribed from the synthetic DNA constructs and electroporated into 293T cells, the cells were monitored daily for NLuc activity and evidence of the cytopathic effects of virus replication. Virus containing medium was collected 6 days post electroporation. B) NLuc activity in 293T cells electroporated with wild type and NLuc modified SARS-CoV-2 RNA transcripts. C) Real-time PCR primer validation using SARS-CoV-2 encoding DNA. NLuc coding sequences were inserted in place of the Orf7a gene. Primers designed to amplify sequences within the DNA constructs confirmed primer specificity. The number of cycles for each gene were normalized to those of the WT virus, or for NLuc, to ΔOrf7a NLuc. For each DNA construct the qPCR data was also normalized to the number of cycles obtained for the N gene (N = 3 technical repeats). D) Real-time qPCR detection of viral RNAs in the medium of infected Vero cells 72 h.p.i. Vero cells were infected with SARS-CoV-2 WT (MOI 0.1) or SARS-CoV-2 ΔOrf7a-NLuc (MOI 1). E) Initial assessment of viral replication in Vero cells. Twenty-four h.p.i. NLuc activity was readily detected in SARS-CoV-2 ΔOrf7a-NLuc infected samples (N = 3 replicate samples). F) NLuc activity in Vero cells infected with wild type SARS-CoV-2 or SARS-CoV-2 ΔOrf7a-NLuc modified virus. NLuc activity was significantly elevated in SARS-CoV-2 ΔOrf7a-NLuc infected cultures across all time points, *** represents p<0.001 in a Students T-Test, N = 3 replicate samples. With prolonged culture NLuc activity within infected samples increased approximately 30-fold.
(TIF)

**S2 Fig. Detection of SARS-CoV-2 virus particles in Vero cells infected with passage 4 virus stocks.** A) After 4 passages of recovered wild type SARS-CoV-2 virus particles in Vero cells, naïve Vero cells were infected and fixed 72 h.p.i. Viral particles could be observed inside and outside of Vero cells indicated by red arrows. Scale bar, 2 μm. B) Higher magnification image of an independent region of the sample in A. Scale bar, 0.5 μm. C) After 4 passages of recovered SARS-CoV-2-ΔOrf7a-NLuc virus particles in Vero cells, naïve Vero cells were infected and fixed 72 h.p.i. Viral particles could be observed inside and outside of Vero cells indicated by blue arrows. Scale bar, 2 μm. Higher magnification image of an independent region of the sample in C. Scale bar, 0.5 μm.
(TIF)

**S3 Fig. Replication assays for different passages of SARS-CoV-2-ΔOrf7a-NLuc virus.** A) The plots show 5000 Vero cells infected with different volumes of virus harvested from repeat passages of the virus (P) in Vero cells. With increasing viral load, there is an increase in background as more NLuc activity is added to each well (Blue). The background signal arises when NLuc, expressed as the virus replicates, is released from infected cells that have lysed upon liberation of the virus particles. N = 2 independent experiments. Similar data were obtained with 2,000 cells and 10,000 cells as shown in **S4 Fig.** B) After subtracting the background noise observed in wells without cells (blue in A), the additional NLuc activity generated through viral replication showed comparable signals for each virus passage. C) The signal-to-noise ratio for each batch was used to identify the optimal conditions for virus replication for subsequent screening experiments. D) Comparison of luminescence signals for virus alone or after incubation with 5000 Vero cells. Luminescence signals are shown relative to the number of PFU for each virus stock. For all panels N = 2 independent experiments, error bars show the standard deviation of repeat measurements of both experiments.
(TIF)

**S4 Fig. SARS-CoV-2-ΔOrf7a-NLuc replication optimization.** NLuc activity at different times after infection with increasing numbers of SARS-CoV-2 virus particles. Enhanced NLuc signals were observed 72 h.p.i. when either A) 2000, B) 5000 or C) 10,000 cells per well were used. Maximal signals were generated with 5000 cells per well, as seeding 10,000 cells tended to reduce NLuc activity. Five thousand cells were used for all subsequent assays.
(TIF)

**S5 Fig. Detection of SARS-CoV-2 infection in lung epithelial cells.** A) Real-time PCR detection of SARS-CoV-2 nucleocapsid (N) RNA in cells 72 h.p.i. Viral RNAs were readily detected in cells infected with WT or ΔOrf7a-NLuc virus particles. N = 3 independent experiments. B) Western blot detection of SARS-CoV-2 nucleocapsid (N) protein in cells 72 hours post infection. Lower levels of the N protein are detected in fibroblasts and lung epithelial cells when compared to 293T and Vero cells suggesting the virus can infect but not replicate in these cell lines. C) Real-time detection of known mediators of SARS-CoV-2 entry into cells. ACE2, NLP1 and TMPRSS2 RNA levels are shown relative to RPLP0. Infectious virus in lung cell lines but no replication.
(TIF)

**S6 Fig. Replication of SARS-CoV-2-ΔOrf7a-NLuc virus in primary lung epithelial cells.** Luciferase counts per second in wells containing virus alone or virus with 5000 primary human lung epithelial cells (CC-2547, Lonza) 72 h.p.i. A small (1.25-fold) increase in signal was observed with 1000 PFU however this was not significant $p > 0.05$, Student's T-test and the Z' factor in these cells was less than 0.1.
(TIF)

**S7 Fig. Effective compounds that suppress SARS-CoV-2 replication with treatment commencing after infection.** A) Schematic for assessing whether any compounds that suppressed virus replication prior to infection could also affect replication post-infection. B) Fifty compounds identified to suppress SARS-CoV-2 replication in **Fig 3** were assessed to identify dose-dependent effects on SARS-CoV-2 replication with treatment progressing after infection (as outlined in A). Nine compounds were found to suppress replication in a dose dependent manner. N = 3 independent experiments, individual data points from each experiment are shown. (TIF)

**S1 Table. Source and growth conditions for cell lines used in this study.**
(XLSX)

**S2 Table. Primers used in this study.**
(XLSX)

**S3 Table. Inhibitors of Sars-CoV-2-ΔOrf7a-Nluc replication in Vero cells.**
(XLSX)

**S4 Table. Effects of identified Inhibitors of Sars-CoV-2-ΔOrf7a-Nluc replication on recombinant NLuc activity.**
(XLSX)

**S5 Table. Compounds that inhibited SARS-CoV-2-ΔOrf7a-Nluc replication in Vero and HUH7 screens.**
(XLSX)

## Acknowledgments

The authors thank Dr. Jennifer Cavet for providing assistance with working at containing level 3 and use of the facilities. The project was approved by the COVID-19 Rapid Response Group, the R3G Research Operations Group and the R3G Executive Committee at the University of Manchester. We would also like to thank staff and students at the University of Manchester for the provision of cell lines for this study: Dr Stuart Cain, Dr Jonathan Humphries, Dr Shiu-wan Chan, Dr Chris Smith, Mrs Rachel Compton, Dr Rogerio Almeida, Dr Sara Gago, Dr Andrew Higham, and Dr Jeremy Herrera. Also thanks to Mrs Nikki-Maria Koudis, Mrs Maryline Fresquet, Dr Bernard Davenport and Dr Richard Naylor from Professor Rachel Lennon's laboratory for help sourcing kidney cell lines and T7 mMessenger mMachine reagents. We would also like to thank Thermo Fisher Scientific for their help in providing microwell plates for this study, with special thanks to Charlotte Connor and Claire Marshall. We would like to thank Miss Anna Pickard for her illustration of the coronavirus.

## Author Contributions

**Conceptualization:** Adam Pickard, Karl E. Kadler.

**Funding acquisition:** Karl E. Kadler.

**Investigation:** Adam Pickard, Ben C. Calverley, Joan Chang, Richa Garva, Sara Gago, Yinhui Lu, Karl E. Kadler.

**Methodology:** Adam Pickard, Ben C. Calverley, Joan Chang.

**Project administration:** Karl E. Kadler.

**Resources:** Sara Gago.

**Supervision:** Adam Pickard, Karl E. Kadler.

**Validation:** Adam Pickard.

**Writing – original draft:** Adam Pickard, Karl E. Kadler.

**Writing – review & editing:** Adam Pickard, Karl E. Kadler.

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
