## [Editor Report · Decision Letter 0]

12 Jun 2021

Dear Prof. Kadler,

Thank you very much for submitting your manuscript "Discovery of re-purposed drugs that slow SARS-CoV-2 replication in human cells" for consideration at PLOS Pathogens. As with all papers reviewed by the journal, your manuscript was reviewed by members of the editorial board and your responses to the REview Commons review were carefully considered. We would like to invite the resubmission of a significantly-revised version that takes into account the reviewers' comments per your responses.

Thank you for using Review Commons and for submitting your manuscript to PLOS Pathogens. Apologies for the delay in handling your manuscript, we were missing critical information that took some time to deliver. Having reviewed your manuscript, the critiques and your response to the critiques, we would be willing to consider a manuscript that was revised according to the responses you have laid out. Please let us know if you have any questions.

We cannot make any decision about publication until we have seen the revised manuscript and your response to the reviewers' comments. Your revised manuscript is also likely to be sent to reviewers for further evaluation.

Sincerely,

Andrew Pekosz, Ph.D.

Section Editor

PLOS Pathogens

Andrew Pekosz

Section Editor

PLOS Pathogens

Kasturi Haldar

Editor-in-Chief

PLOS Pathogens

orcid.org/0000-0001-5065-158X

Michael Malim

Editor-in-Chief

PLOS Pathogens

orcid.org/0000-0002-7699-2064

Thank you for using Review Commons and for submitting your manuscript to PLOS Pathogens. Apologies for the delay in handling your manuscript, we were missing critical information that took some time to deliver. Having reviewed your manuscript, the critiques and your response to the critiques, we would be willing to consider a manuscript that was revised according to the responses you have laid out. Please let us know if you have any questions.
---

## [Editor Report · Decision Letter 1]

26 Jul 2021

Dear Prof. Kadler,

We are pleased to inform you that your manuscript 'Discovery of re-purposed drugs that slow SARS-CoV-2 replication in human cells' has been provisionally accepted for publication in PLOS Pathogens.

Best regards,

Andrew Pekosz, Ph.D.

Section Editor

PLOS Pathogens

Andrew Pekosz

Section Editor

PLOS Pathogens

Kasturi Haldar

Editor-in-Chief

PLOS Pathogens

orcid.org/0000-0001-5065-158X

Michael Malim

Editor-in-Chief

PLOS Pathogens

orcid.org/0000-0002-7699-2064
---

## [Editor Report · Acceptance letter]

10 Aug 2021

Dear Prof. Kadler,

We are delighted to inform you that your manuscript, "Discovery of re-purposed drugs that slow SARS-CoV-2 replication in human cells," has been formally accepted for publication in PLOS Pathogens.

Best regards,

Kasturi Haldar

Editor-in-Chief

PLOS Pathogens

orcid.org/0000-0001-5065-158X

Michael Malim

Editor-in-Chief

PLOS Pathogens

orcid.org/0000-0002-7699-2064